# Effects of Workplace Ostracism on Burnout among Nursing Staff during the COVID-19 Pandemic, Mediated by Emotional Labor

**DOI:** 10.3390/ijerph20054208

**Published:** 2023-02-27

**Authors:** Feng-Hua Yang, Shih-Lin Tan

**Affiliations:** 1Department of International Business Management, Da-Yeh University, Changhua County 515006, Taiwan; 2Ph.D. Program in Management, Da-Yeh University, Changhua County 515006, Taiwan

**Keywords:** workplace ostracism, emotional labor, emotional labor, COVID-19

## Abstract

This study investigated the effects of workplace ostracism on emotional labor and burnout among current nursing staff during the COVID-19 pandemic, as well as the relationship between the surface acting and deep acting of emotional labor as the mediators of workplace ostracism and burnout. The sample for this study consisted of 250 nursing staff recruited from Taiwanese medical institutions, and the questionnaire was divided into two stages. The first stage included questions about ostracism and personal data, and then two months later the same respondents completed part two of the questionnaire regarding emotional labor and burnout, which solved the problem of common-method variance (CMV). The results of this study indicate that ostracism had a positive and significant effect on burnout and surface acting, but its negative effect on deep acting was not supported. While surface acting showed partial mediation between ostracism and burnout, deep acting did not have a significant mediating effect between ostracism and burnout. These results can provide a reference for practice and researchers.

## 1. Introduction

Between 2019 and 2022, the world has been in the grip of the COVID-19 pandemic. The most important resources in pandemic preparedness—nursing staff—are in the first line facing the pandemic, providing care for the infected, and giving necessary medical support; thus, the demand for nurses and their workload have increased, which has caused great psychological stress. Studies have shown that due to the COVID-19 pandemic there has been a significant increase in nurses’ turnover intention, and that the negative impacts on nursing staff’s mental health should be used to predict nurses’ turnover intention [1]. During the COVID-19 pandemic, fear, panic, and misinformation about the pandemic drove violent attacks and abuse against nursing staff [2]. Healthcare workers also experience a high rate of infection during pandemics, which leads to significant psychological stress [3]. Based on the above premises, workplace relationships in healthcare settings are a source of stress that should be explored.

A piece of literature published in Taiwan discussed how job stresses caused by behavioral and verbal violence, as experienced by nursing staff, are also major reasons for nursing staff leaving the workplace [4]. It has been suggested that burnout and emotional exhaustion caused by job stress are the main factors that trigger a sense of loss of self and contribute to nursing staff turnover [5]. Therefore, this study found that the ostracism experienced by nursing staff in the work environment is one of their major sources of job stress.

Burnout can occur due to stress from the job, role, and organization, as well as from the individual [6]. An examination of the effects of the three components of job burnout—namely, personality traits, job stress, and stress coping strategies—found that personality traits had a negative correlation with job burnout, while job stress had a positive correlation with job burnout [7]. According to the literature, a study of the turnover intention of physical education teachers in public high schools in the United States found that teachers’ burnout fully mediated emotional labor and turnover intention; therefore, it was concluded that personality traits, job stress, career and life factors, and emotional labor had significant effects on job burnout [8].

In 1997, Williams and Sommer (1997) [9] found that ostracism is the self-perception of being ignored or rejected by others or groups, and its presence affects all activities of human life. Ferris et al. (2008) [10] defined workplace ostracism as employees being ignored by others or by the organization, and many studies have confirmed that it affects employees’ sense of belonging to the organization. Therefore, in the workplace environment, ostracism affects work engagement and generates negative emotions and behaviors.

Many scholars have studied job burnout caused by workplace ostracism. Relevant studies found that (a) workplace ostracism was positively related to job burnout; (b) the relationship depended on job autonomy, and the relationship was weaker when job autonomy was higher; and (c) the relationship depended on employees’ future time perspective orientation, and the relationship was weaker for employees with higher future time perspective orientation [11]. Thus, while this study found that workplace ostracism could contribute to job burnout and stress, there was still a missing quantifiable intermediate variable between workplace ostracism and job burnout that contributes to employee burnout and stress at work.

The emotional labor theory points out that different employees use different combinations of emotional regulation strategies to manage their emotional expression at work [12]. Some scholars have suggested that the theoretical approach to emotional labor revolves around three themes: internal states, internal processes, and external behavioral displays (Glomb and Tews, 2004) [13], which suggests that emotional labor has a mediating effect when employees have emotional reactions at work and, thus, it can be used to explore the relationship with burnout. Then, we can conclude that personality traits, work stress, occupational and life factors, and emotional labor have a significant impact on job burnout. Therefore, emotional labor is a possible factor in the medical environment.

The objective of this study was to analyze and summarize the current knowledge on factors/potential factors contributing to burnout among nursing staff amidst the COVID-19 pandemic. While most health professionals have had to shoulder the burden, nursing staff are not often recognized as being vulnerable; hence, little attention is paid to workplace ostracism and emotional labor within this group. This study also makes a few recommendations on how best to prepare intervention programs for nursing staff.

## 2. Theoretical Background

### 2.1. Workplace Ostracism

Ostracism has a negative and debilitating effect on human self-esteem, sense of belonging, control, and the need for a meaningful existence [14]. Hitlan et al. (2006, 2009) [15,16] found that verbal ostracism affected work attitudes and behaviors, and they developed the WOE scale. Robinson et al. (2013) [17] collated models of workplace ostracism and proposed clear concepts for workplace ostracism, such as organizational shunning, social exclusion, and rejection. Lin (2017) [8] integrated the definitions of workplace ostracism by scholars and summarized the following two characteristics: First, workplace ostracism is a subtle and hidden negative interpersonal behavior, which is a kind of cold violence in the workplace and can have far-reaching effects on employees and organizations. Second, the source of ostracism comes from the people encountered in the workplace, including supervisors, coworkers, and subordinates.

### 2.2. Burnout

Burnout has three conceptualized components: one is emotional exhaustion, the second is depersonalization, and the third is reduced personal accomplishment [18]. In 1982, Maslach described burnout as a syndrome of emotional exhaustion and cynicism and found that it often occurred in individuals engaged in interpersonal work. Stevenson (1994) [19] described burnout as a process of gradual exhaustion and debilitation, a failure of employees to be motivated by work stress, a sense of loss, and a loss of physical and psychological capacity to contribute to the organization. Lee and Ashforth (1996) [20] applied secondary data analysis and found that job demand was more important than a lack of job resources and was a predictor of job burnout. In a study of primary care clinicians in Catalonia, Seda-Gombau et al. (2021) [21] found that emotional exhaustion jumped from 55% to 77% due to pandemic-related burnout.

The frontline health workforce is experiencing a high workload and multiple psychosocial stressors that may affect their mental and emotional health, leading to burnout symptoms [22]. Recent studies generally show that the introduction of COVID-19 has heightened existing challenges that medical personnel face, such as increasing workload, which is directly correlated with increased burnout [23]. Thus, paying attention to mental health issues, reducing the workload of medical personnel through adjusting their work shifts, reducing job-related stressors, and creating a healthy work environment may prevent or reduce burnout [24].

### 2.3. Emotional Labor

The psychological process of the emotional labor journey was divided by Hochschild (1983) [25] into surface acting, deep acting, and diversity. Surface acting refers to disguising the individual’s inner emotions, abandoning the original expression of emotions, and expressing “appropriate” emotions according to the company’s expression principles. Deep acting refers to the individual’s use of control for their inner thoughts, meaning that they transform them into real emotional experiences that truly reflect the emotions required by the company, i.e., real emotions. A study mentioned that emotional labor does not necessarily present negative outcomes if employees have sufficient coping resources to protect against the negative effects of emotional labor [26]. Hochschild divided emotional labor into two models: deep acting and surface acting. Surface acting refers to employees’ use of emotions for expression that are consistent with the organization’s expectations; meanwhile, in deep acting, employees express the emotions expected by the organization and adjust their true emotions to be internally consistent. Based on the above premises, this study investigated the effect of emotional labor as a mediator between workplace ostracism and burnout; when employees suffer from workplace ostracism, their burnout varies according to their attitudes toward their emotional performance.

### 2.4. Hypothesis

The conceptual framework is presented in Figure 1.

#### 2.4.1. Workplace Ostracism and Job Burnout

Qian et al. (2017) [11] suggested that a priority for management to manage employee burnout is to take measures that reduce the occurrence of workplace ostracism. Empirical studies have shown that emotional exhaustion occurs when employees are ostracized in the workplace [27]. This suggests that employees will try to adjust their attitudes and behaviors in the face of workplace ostracism and spend more time dealing with interpersonal relationships. In severe cases, they will choose avoidance and experience burnout because they cannot adjust. Due to the impact of the COVID-19 pandemic, there has been a significant phenomenon of increased turnover intentions among primary care clinicians due to job burnout [28]. Thus, based on the above analysis, this study proposes the following hypothesis:

**Hypothesis** **1** **(H1).***Workplace ostracism is positively related to job burnout*.

#### 2.4.2. Workplace Ostracism and Emotional Labor

Previous research studied the association between emotional labor, job burnout, and job satisfaction among schoolteachers in China and found a correlation between job burnout and emotional labor, as hypothesized, with a positive correlation between superficial behaviors and job burnout, and a negative correlation between deep behaviors and a lack of personal fulfillment [29]. Emotional labor is prevalent in nursing, and a great deal of emotional labor is involved in caring for patients, while at the same time workplace ostracism creates more job stress [30]. Based on the above analysis, this study proposes the following hypothesis:

**Hypothesis** **2** **(H2).***Workplace ostracism has a positive effect on surface acting*.

When nursing staff face workplace ostracism, they are less able to adjust their internal feelings through self-perception assessment. Based on the above analysis, this study proposes the following hypothesis:

**Hypothesis** **3** **(H3).***Workplace ostracism has a negative effect on deep acting*.

#### 2.4.3. Emotional Labor as a Mediator between Workplace Ostracism and Job Burnout

Affective events theory emphasizes the process that follows an employee’s emotional response in the workplace, focusing entirely on the process of personal judgment [31,32]. AET shows that when employees face the negative events of workplace ostracism, they will abandon their own emotional expressions and show “appropriate” emotions according to the expression principles required by the company, meaning that their surface acting will be enhanced, which may increase employees’ job stress and lead to an increase in job burnout. Based on the above analysis, this study proposes the following hypothesis:

**Hypothesis** **4** **(H4).***Surface acting will mediate the relationship between workplace ostracism and job burnout, i.e., when workplace ostracism increases, surface acting will increase which, in turn, will increase job burnout*.

Conversely, when employees face workplace ostracism, they are less able to adjust their internal feelings through self-perception assessment. Therefore, deep acting will decrease, while employees’ job stress will increase, which will lead to an increase in job burnout. Based on the above analysis, this study proposes the following hypothesis:

**Hypothesis** **5** **(H5).**
*Deep acting will mediate the relationship between workplace ostracism and job burnout, i.e., when workplace ostracism increases, deep acting will decrease which, in turn, will increase job burnout.*


## 3. Methods

### 3.1. Participants and Procedure

This study enrolled the current nursing staff in medical institutions in Taiwan as subjects, meaning that these staff work in medical centers, regional hospitals, district hospitals, clinics, and long-term nursing organizations. To avoid common-method variance, we collected data at two time points for the same participants, so we had not calculated the sample size. A total of 262 volunteer nursing staff were recruited via professional nursing associations and were numbered. As the information was quickly categorized by the criteria when receiving and processing information, different information might be categorized as one category, which could easily lead to the problem of common-method variance (CMV). In order to avoid the validity of this study being affected by the CMV factor, the questionnaires were administered in a time-isolated manner, with IV and DV measured at different times. The questionnaires were designed in two stages: the data regarding ostracism and personal data were obtained in the first stage (T1), while the data on emotional labor and burnout were obtained in the second stage (T2) 2 months later. This study obtained 250 valid questionnaires (95%) after invalid questionnaires in both stages were removed. Personal information was repeatedly obtained to improve the consistency of the two testing stages for the same sample. The participants’ demographic information is presented in Table 1.

### 3.2. Measurement

The surveys were administered in Mandarin. To ensure the content equivalence of the translated version, Chinese speakers who were proficient in English were employed to scrutinize all of the scales in this study following a back-translation procedure (Brislin, 1980) [33]. All participants responded to survey items using a seven-point scale, ranging from 1 “very strongly disagree” to 7 “very strongly agree”.

#### 3.2.1. Burnout

We measured burnout using the Maslach Burnout Inventory–General Survey [34]. This questionnaire has been repeatedly validated by international academics and has high reliability and validity. The 16-item scale comprises three dimensions: exhaustion (5 items), cynicism (5 items), and professional efficacy (6 items). The sample items are “e.g., I feel burned out from my work” (exhaustion), “e.g., I have become less enthusiastic about my work” (cynicism), and “e.g., I feel confident that I am effective at getting things done” (professional efficacy). The Cronbach’s alpha values in this study were 0.882, 0.854, and 0.843, respectively.

#### 3.2.2. Ostracism

We measured ostracism using the Workplace Ostracism Scale [10]. This questionnaire has been repeatedly validated by international academics and has high reliability and validity. The sample item is “e.g., Others ignored you at work”. Cronbach’s alpha in this study was 0.912.

#### 3.2.3. Emotional Labor

We measured emotional labor using the Brotheridge and Lee (2003) [35] ELS study questionnaire—a 6-item questionnaire with the 2 dimensions of surface acting and deep acting. This questionnaire has been repeatedly validated by international academics and has high reliability and validity. The sample items are “Hide my true feelings about a situation” (surface acting) and “I try to feel the emotions that I have to show in my work” (deep acting). The Cronbach’s alpha values in this study were 0.823 and 0.838, respectively.

## 4. Result

### 4.1. Confirmatory Factor Analysis

This study used AMOS to conduct confirmatory factor analysis (CFA), which can be used to deal with the covariance between the observed variables and their potential variables, and the valid questionnaires were used to test whether all of the possible items could be assigned to the theoretically expected variables. The collected data were tested for common-method bias using Harman’s single-factor test, which was below the 40% threshold, showing that the data did not suffer from serious common-method bias. A higher CR (composite reliability) value of a dimension indicates higher internal consistency, and the value should be at least 0.60 [36]. The average variance extracted (AVE) is the ratio of the variance explained by potential variables and is an indicator of convergent validity; thus, the value must reach 0.50 [37]. Jöreskog and Sörbom (1989) [38] suggested that items with too low factor loadings should be deleted after standardization, and that the factor loading value should be above 0.45. The tests were conducted in this study according to the abovementioned scholars’ recommendations for each indicator.

In this study, the factor loadings for each of the six research dimensions were tested with the CFA test model and were all above 0.45. The composite reliability and AVE were above the recommended cutoff points, except for professional efficacy, where the AVE was 0.481. The convergence of this study was acceptable, as the overall dimension model was significant for the question items, and the internal consistency was also significant.

This study tested the six research dimensions for model fit with CFA: CMIN/DF = 2.988, which met the model fit indices (CMIN = 1341.830, df = 449, *p* = 0.00), GFI = 0.750, AGFI = 0.706, NFI = 0.746, IFI = 0.816 CFI = 0.750, RMR = 0.138, and all of the above indices met the validation criteria. PNF and PCFI were both greater than 0.5, which indicates that the research dimensions were not complex. The values of variances and residuals were all positive and significant; thus, the research dimensions were single latent variables, and second-order CFA model analysis was not required.

#### 4.1.1. Ostracism

We tested the measurement model using the Workplace Ostracism Scale and found no items with factor loadings lower than 0.45, with the highest being 0.856 and the lowest being 0.587. The *t*-values for the observed variables were all significant, with a maximum of 12.507 and a minimum of 8.777. The CR of 0.923 and AVE of 0.548 both met the criteria.

#### 4.1.2. Surface Acting

We tested the measurement model using the Surface Acting Scale and found no items with factor loadings lower than 0.45, with the highest being 0.818 and the lowest being 0.731. The highest *t*-values of 11.386 and the lowest of 10.796 were observed for the observed variables, both of which reached significance. The CR of 0.824 and AVE of 0.610 both met the criteria.

#### 4.1.3. Deep Acting

We tested the measurement model using the Deep Acting Scale and found no items with factor loadings lower than 0.45, with the highest being 0.878 and the lowest being 0.644. The highest *t*-values of 10.647 and the lowest of 10.644 were significant for each observed variable. The CR of 0.846 and AVE of 0.651 both met the criteria.

#### 4.1.4. Exhaustion

We tested the measurement model using the Exhaustion Scale and found no items with factor loadings lower than 0.45, with the highest being 0.860 and the lowest being 0.715. The highest and lowest *t*-values of 13.171 and 10.943, respectively, were found to be significant for all observed variables. The CR of 0.883 and AVE of 0.602 both met the criteria.

#### 4.1.5. Cynicism

We tested the measurement model using the Cynicism Scale and found no items with factor loadings lower than 0.45, with the highest being 0.813 and the lowest being 0.633. The highest and lowest *t*-values of 12.987 and 9.871, respectively, were observed for each variable, and both of which reached significance. The CR of 0.855 and AVE of 0.543 both met the criteria.

#### 4.1.6. Professional Efficacy

We tested the measurement model using the Professional Efficacy Scale and found no items with factor loadings lower than 0.45, with the highest being 0.792 and the lowest being 0.606. The *t*-values for the observed variables ranged from the highest value of 9.673 to the lowest value of 7.960, and both were significant. The CR of 0.846 met the criterion, while the AVE of 0.481 was slightly lower than the criterion of 0.50. The convergence of this study was acceptable, as the overall dimension model was significant for the question items, and the internal consistency was also significant.

### 4.2. Regression Analysis

Our hypotheses were tested using hierarchical multiple regression analysis with a three-step procedure, and we also used 95% bias-corrected confidence intervals to estimate the statistical significance of this mediated effect. As shown in Table 2, the β value of workplace ostracism to job burnout was 0.214, which shows a significant positive correlation; thus, H1 was supported. Therefore, when workplace ostracism was encountered in an unfriendly workplace environment, it could cause an increase in job burnout. The β value of workplace ostracism alone on surface acting was 0.136, which shows a significant positive correlation; thus, H2 was supported. Therefore, emotional labor was positively influenced by workplace ostracism, and caregivers would use a lot of surface acting to hide their true emotions when caring for patients. The β value of workplace ostracism on deep acting was −0.046, which did not show a significant negative correlation; thus, H3 was not supported, meaning that caregivers were unable to adapt through an internal cognitive adjustment when facing workplace ostracism. The relationship between workplace ostracism and burnout was mediated by surface acting, and the β value was 0.280 with a significant positive correlation; thus, H4 was supported, meaning that when workplace ostracism increased, surface acting also increased which, in turn, increased burnout. The relationship between workplace ostracism and burnout was not mediated by deep acting, with a β value of −0.108; thus, H5 was not supported, meaning that caregivers were unable to adapt through internal cognitive adjustment and reduce burnout when facing workplace ostracism.

## 5. Discussion

This study examined the impact of workplace ostracism on job burnout among current nursing staff in healthcare facilities during the COVID-19 pandemic and how emotional labor mediates the relationship between the two. Five research hypotheses were developed based on the research concepts and according to the literature review.

H1 verified that workplace ostracism has a positive and significant effect on job burnout, which is consistent with the literature [11,27], meaning that the higher the level of workplace ostracism experienced by caregivers under the stress of the pandemic, the more it causes job burnout and emotional exhaustion.

H2 verified that workplace ostracism has a significant positive effect on surface acting, which is consistent with the literature [30], meaning that when caregivers are under the stress of a pandemic, the higher the level of workplace ostracism they experience, the more they mask their true emotions, which is consistent with the expectation that caregivers should present superficial attitudes in the healthcare environment, thereby effectively reducing tension between patients and doctors.

H3 was not supported, which stated that workplace ostracism has a negative effect on deep acting, which is significantly different from the hypothesis and results of Li (2018) [30] on workplace ostracism and deep acting among caregivers, whose study hypothesized a positive relationship between workplace ostracism and deep acting and, thus, the findings were not supported. We believe that it may be the professional rationality of nursing work that prevents emotional labor from affecting their professional judgment, especially when faced with irrational behaviors of patients, i.e., nursing staff will not have emotional reactions to patients because of the principle of medical priority.

H4 was supported, verifying that when workplace ostracism increased, the mediating relationship with surface acting also increased, but job burnout decreased. The results show that surface acting has a mediating effect on workplace ostracism and job burnout. When nursing staff faced workplace ostracism, they concealed their true feelings and showed emotions that were consistent with the doctor–patient relationship. Therefore, the frequency of surface acting increased because the authority of the nursing profession is less. This was also affected by the patient’s emotions.

H5 verified that deep acting provided no mediating effect between workplace ostracism and job burnout, meaning that the findings did not support the idea that nursing staff would adjust their inner feelings through deep acting when facing a workplace ostracism situation, which would lead to increased job stress and job burnout.

## 6. Conclusions

The results of this study show that workplace ostracism did contribute to nursing staff’s burnout during the COVID-19 pandemic, which is consistent with the studies of Heaphy and Dutton (2008) [39] and Jiang et al. (2020) [27]. In addition, surface acting in the emotional labor theory is similar to Cheung et al.’s (2011) [29] findings regarding the association between emotional labor and burnout, which showed a positive correlation between superficial behavior and job burnout. This suggests that nursing staff use surface acting to present attitudes that meet the expectations of the hospital and patients and reduce their true responses, even during situations of workplace ostracism.

Kwon et al. (2021) [40] examined the positive correlation between nursing staff’s emotional labor, job burnout, and the resulting job errors, which was consistent with the results of this study. This means that workplace ostracism is associated with increased surface acting behaviors and contributes to increased job burnout. It was inferred that job burnout may affect job accuracy—especially during a pandemic, when the volume of patient visits is increased and nursing staff must deal with a large number of patient relationships, which makes them become more fatigued at work and neglect proper work processes.

## 7. Limitations and Future Studies

A limitation of this study is that the participants were active frontline nursing staff in medical institutions, but pandemic measures vary by region, which means that the results may not be applicable to other regions where the pandemic is less severe or where medical resources are more available. In this study, a two-stage mediation analysis method was used to conduct research on the participants in line with the research purpose. There are two possible reasons why this study did not obtain support for a significant negative effect of deep acting on the association between workplace ostracism and burnout: First, that the nursing staff did not understand the significance of deep acting as an assessment of self-perception to adjust their true feelings in the doctor–patient relationship; thus, future research could explore the mediating effect of deep acting by making the questions more precise and easier to understand. For example, “I will try to truly experience the emotions that I have to show to others” should be revised to read “I will try to change my mind about the emotions I show in order to facilitate the doctor-patient relationship”. In addition, most of our participants were frontline clinical nursing staff who were preoccupied with their job duties and pandemic prevention and did not have the capacity or time to maintain the doctor–patient relationship and perform some of the behaviors described in this study. Caregiving is not a service industry that requires high levels of customer satisfaction and repurchase support; therefore, it is suggested that future researchers should address these limitations, which may yield different results than this study.

## Figures and Tables

**Figure 1 ijerph-20-04208-f001:**
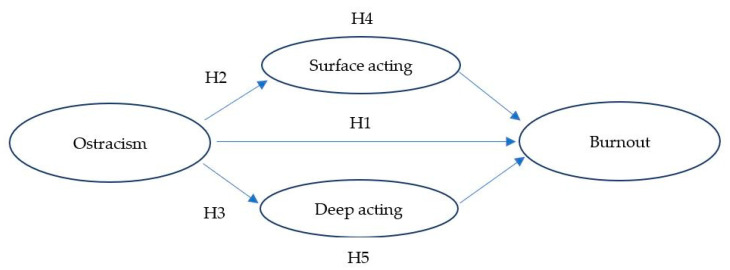
Hypothesized Model.

**Table 1 ijerph-20-04208-t001:** Participants’ demographic characteristics.

Characteristics		n	%
Gender	Male	4	1.6%
	Female	246	98.4%
Age (years)	20–29	29	11.6%
	30–39	85	34%
	40–49	109	43.6%
	50–59	21	8.4%
	60 and above	6	2.4%
Educational level	Junior high school	0	0%
	Senior high school	14	5.6%
	University	224	89.6%
	Master’s and doctorate	12	4.8%
Supervisor	Yes	30	12%
	No	220	88%
Tenure (years)	5 or less	29	8%
	6–10	40	16%
	11–20	124	49.6%
	21–30	51	20.4%
	31 or over	6	2.4%
Institution served	Medical center	33	13.2%
	Regional hospital	67	26.8%
	District hospital	30	12%
	Clinic	29	11.6%
	Long-term nursing organization	91	36.4%
Total		250	100%

**Table 2 ijerph-20-04208-t002:** Results of regression analysis.

	**Step 1**				
	**Burnout (DV)**				
	**β**	**R^2^**	**ΔR^2^**	**F**	** *t* **
Ostracism	0.214 **	0.046	0.042	11.929	3.454
Surface acting	0.136 *	0.019	0.015	4.686	2.165
Deep acting	−0.046	0.002	−0.002	0.537	−0.733
	**Step 2**				
	**Burnout (DV)**				
	**β**	**R^2^**	**ΔR^2^**	**F**	** *t* **
IV Ostracism	0.176 **				2.928
Mediator Surface acting	0.280 ***	0.123	0.116	17.278	4.651
IV ostracism	0.209 **				3.383
Mediator Deep acting	−0.108	0.058	0.050	7.550	−1.752

*** = *p* < 0.00; ** = *p* < 0.01; * = *p* < 0.05.

## Data Availability

The data presented in this study are available upon request from the corresponding author.

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
