# Peer review of "Effects of Workplace Ostracism on Burnout among Nursing Staff during the COVID-19 Pandemic, Mediated by Emotional Labor"

_ijerph, 2023, doi:10.3390/ijerph20054208_

Round 1

Reviewer 1 Report

Thanks to the authors for sharing their manuscript. I think that the authors have conducted an interesting and important scientific study, but I would like to make a few comments before recommending that the manuscript be published:

·        I would like to understand why the authors chose mediation analysis over regression analysis in a two-stage study. Although mediation analysis can be found even in cross-sectional studies, I think this point should be highlighted in the manuscript.

·        It remains unclear from the text of the manuscript whether the authors calculated the sample size? I would add this information to the Method.

·        When describing the measures, the authors refer to the sources in which the original instruments were published. I would advise the authors to indicate in which language the questionnaire was given to the respondents. If it is not English, they should either refer to the adapted versions of the measures or describe the results of confirmatory factor analysis in addition to the values of Cronbach's alpha coefficients.

·        If the authors justify the use of mediation analysis in a two-stage study, I invite them to highlight this decision in the limitations and future research perspectives.

·        Stylistic point: it seems to me that the text would benefit if too many introductory words were excluded from it (‘however’, ‘clearly’, ‘moreover’, ‘similarly’, etc.).

·        Please note the References: DOIs are marked as individual sources, which doubles the list of references.

Author Response

Dear Reviewer,

The revised as attached

Reviewer 2 Report

I would like to start by show my appreciation for the effort done by the authors. The paper is well written and generally follows the structure recommended by the journal. 

Nevertheless, I would like to share a few comments that I expect may contribute to further improve the quality of the paper:

- Introduction: be more clear and compelling about the novelty of this research; in what way is it different from previous research? what is the added value of this research? what gap does it try to address?

- Please be more clear about the relevance of ostracism as an antecedent of burnout.

- Although your paper is about burnout of healthcare workers during Covid-19, your theoretical background uses just a few references about the topic; please reference more published work on this topic

- Discussion: you should discuss deeper why H3 is not supported

- References: you need to correct the references; several links are presented as references (numbered); after that, correct the references numbers in the text

Author Response

Dear Reviewer,

The revised as attached.
